# Successful Pre-Rewarming Resuscitation after Cardiac Arrest in Severe Hypothermia: A Retrospective Cohort Study from the International Hypothermia Registry

**DOI:** 10.3390/ijerph19074059

**Published:** 2022-03-29

**Authors:** Evelien Cools, Marie Meyer, Delphine Courvoisier, Beat Walpoth

**Affiliations:** 1Department of Acute Medicine, Division of Anaesthesiology, University Hospitals Geneva, 1205 Geneva, Switzerland; 2Department of Anaesthesiology, Centre Hospitalier University Vaudois, 1011 Lausanne, Switzerland; meyer.marie1@gmail.com; 3Quality of Care Unit, University Hospitals of Geneva, 1205 Geneva, Switzerland; delphine.courvoisier@hcuge.ch; 4Emeritus, Department of Cardiovascular Surgery, University Hospitals of Geneva, 1205 Geneva, Switzerland; beat.walpoth@unige.ch

**Keywords:** accidental hypothermia, resuscitation, rewarming, mountain medicine

## Abstract

The aim of our study is to investigate successful pre-rewarming resuscitation after hypothermic cardiac arrest (HCA). The hypothermic heart may be insensitive to defibrillation when core temperature is below 30 °C and after successful defibrillation, sinus rhythm often returns into ventricular fibrillation. Recurrent defibrillation attempts may induce myocardial injury. Discrepancy exists concerning pre-rewarming defibrillation between the guidelines of the European Resuscitation Council and American Heart Association. The International Hypothermia Registry (IHR) gathers hypothermia cases. The primary outcome was survival. Secondary outcomes were the characteristics of defibrillation, the effect of Adrenaline administration under 30 °C, and the duration of CPR. Of the 239 patients, eighty-eight were in cardiac arrest at arrival of the rescue team. Successful pre-rewarming resuscitation was obtained in 14 patients. The outcome showed: seven deaths, one vegetative state, two patients with reversible damage, and four patients with full recovery. A total of five patients had a shockable rhythm, and defibrillation was successful in four patients. The response rate to Adrenaline was reported as normal in six patients. There were no statistically significant differences in the presence of a shockable rhythm, the success of defibrillation, and the effect on Adrenaline administration between the survivors and non-survivors. Successful resuscitation in severe hypothermia is possible before active rewarming and arrival in the hospital, thus improving the chance of survival.

## 1. Introduction

Hypothermia (core temperature ≤ 35 °C) is a frequent and life-threatening problem after mountain accidents, near-drowning, and intoxications [1,2], and can provoke arrhythmia, reduced cardiac contractility, and cardiac arrest [3,4,5,6]. The hypothermic heart may be insensitive to defibrillation when core temperature does not reach 30 °C. Sinus rhythm often returns into ventricular fibrillation. Repeated defibrillation can induce myocardial injury [7]. The ERC guidelines suggest delaying further defibrillation attempts until the core temperature is >30 °C if ventricular fibrillation persists after three shocks. Adrenaline should be withheld if core temperature is <30 °C [7]. ACLS guidelines define that it may be reasonable to perform further defibrillation attempts according to the standard BLS algorithm and to consider administration of a vasopressor during cardiac arrest as cited in the standard ACLS algorithm concurrent with rewarming strategies [8]. This discrepancy can be explained by the different interpretations of mainly animal data, which show that vasopressors increase the chances of successful defibrillation < 30 °C. However, the return of spontaneous circulation (ROSC) is not stable and tends to degrade into VF again [9,10].

The aim of this study was to evaluate successful pre-rewarming resuscitation after hypothermic cardiac arrest (HCA) and the ICU outcome of these patients. Secondary outcomes were the characteristics of defibrillation, Adrenaline administration in patients with a core temperature under 30 °C, and the duration of CPR.

## 2. Materials and Methods

This study is a retrospective cohort analysis of data from the International Hypothermia Registry (https://www.hypothermia-registry.org (accessed on 1 December 2021).) (IHR). In 2008, we created the IHR to gather information of patients who were victims of accidental hypothermia. The International Hypothermia Registry contains information of patients of all ages diagnosed with accidental hypothermia and a core temperature less than 32 °C. There are no restrictions on the etiology of hypothermia. This registry is mainly prospective but retrospective entries are welcome. A total of fifty-two hospitals in Austria, France, Italy, Poland, Switzerland, the United Kingdom, and the United States of America participate in the IHR. The majority of the centers are university hospitals, and a minor number are primary or secondary hospitals, located in the mountains. One year after the accident, patients were contacted and outcome data were investigated.

For this analysis, we used patient data entered in the IHR until the 30 November 2021. Ethical approval was obtained from the ethical committee of Geneva, Switzerland (No 08-040R).

An extraction of the database was performed by D.C. on the 1 December 2021. We selected all patients who were in cardiac arrest before hospital admission. We considered the patients with a return of spontaneous circulation before rewarming. These patients were extracted. The demographical data observed were: gender, age, and the year and country of the accident. Prehospital data were core temperature at arrival of the rescue team, the presence of a witnessed cardiac arrest, asphyxia, the accident mechanism, the initial heart rhythm, the duration of CPR, and the rhythm after ROSC. In addition, the number of successful defibrillation attempts and the reaction on Adrenaline administration before rewarming were recorded. Hospital data were the pH, potassium, and lactate before rewarming (the first analysis available after arrival in the hospital), the rewarming methods, and the complications during rewarming. The ICU outcome was registered.

Deep accidental hypothermia is rare. The low number of cases can result in significant bias. Some cases were entered retrospectively, which can result in recall bias. Confounding factors such as the severity of trauma can influence the final outcome. The description of the reaction on Adrenaline can be subjective.

### Statistical Analysis

All data were analyzed as number of cases and percentages (%), median and inter-quartile range (IQR) or mean values, and standard deviations (SD) for continuous variables as appropriate. Groups were compared using Fisher exact test for categorical variables and Wilcoxon rank sum test for continuous variables. The level of significance was set at *p* < 0.05. We used deletion methods to eliminate missing data.

## 3. Results

### 3.1. Demographics and Survival

Of the 239 patients registered in the IHR, eighty-eight (37%) were in cardiac arrest at arrival of the rescue team. Successful pre-rewarming resuscitation with return of spontaneous circulation (ROSC) was obtained in 14 (16%) patients (Figure 1). The 14 patients of our study came from six different hospitals in four different European countries (Table 1). Accidents occurred between 1994 and 2015 and cases were entered between 2011 and 2016. Patient characteristics are mentioned in Table 1. Mean age was 33.6 years, 78% were men, with a mean core temperature of 26.9 +/− 1.6 °C. Suspicion of asphyxia was found in 10 cases and eight patients had an unwitnessed cardiac arrest. Regarding demographics, there were no significant statistical differences between the survival and non-survival group for age, core temperature, witnessed cardiac arrest, or asphyxia (Table 2).

### 3.2. Patient Outcome

The ICU outcome showed: seven deaths, one vegetative state, two patients with reversible damage, and four patients with full recovery (see Figure 1). Death was caused by pulmonary edema in two patients, brain death in four patients, and a persistent neurologic central deficit (GCS3) in one patient (Table 3).

There were no statistically significant differences in potassium and lactate levels between the survivors and non-survivors (Table 2). A total of three patients were rewarmed externally (ECMO stand-by in one patient), one body cavity lavage, three invasive CoolGard combined with external rewarming, three invasive CoolGard, two ECMO, and unknown in one patient. In the one patient with body cavity lavage, rewarming was switched to V-V- ECMO due to acute lung failure.

### 3.3. Defibrillation Attempts and Response Rate

The first heart rhythm before rewarming was pulseless electrical activity (PEA) in six patients, asystole in three, and ventricular fibrillation (VF) in five patients. Of the patients, five had a non-shockable rhythm at arrival of the rescue team, but had a successful resuscitation. Defibrillation before re-warming was performed in five patients. Defibrillation was successful in four patients (Table 4). The number of shocks ranged from one to four (number of attempts unknown in one patient). In one patient, only one defibrillation attempt was performed, which was unsuccessful. There were no statistically significant differences in the presence of a shockable rhythm, the success of defibrillation, and the reaction on Adrenaline administration between the survivors and non-survivors (Table 2). A total of four patients with a non-shockable rhythm (PEA) and three patients in ventricular fibrillation survived. The energy level for defibrillation was only mentioned in one patient (300 J—patient 9).

Heart rhythm after successful resuscitation and defibrillation was sinus rhythm in 10, atrial fibrillation (AF) in two, and unmentioned in two patients. The electrocardiogram was not recorded in these two patients, but blood pressure was more than 80/40 mmHg before rewarming.

### 3.4. Cardiovascular Drugs Administered and Response Rate

Adrenaline was administered in eight patients with a core temperature under 30 °C, and a normal response (i.e., the same reaction as in a non-hypothermic patient) was reported in six patients (see Table 4).

### 3.5. Duration of CPR

The duration of CPR was known in 10 of the 14 patients (71%) with successful pre-rewarming resuscitation (Table 4). There was no statistically significant difference in CPR duration between the survivors and non-survivors (Table 2). However, the CPR duration was half (12′ vs. 29′) in the survivors and this difference reached almost significance.

## 4. Discussion

Successful resuscitation in severe hypothermia is possible before active rewarming. In our study, 14 of 88 patients in cardiac arrest attained ROSC before rewarming. Despite the successful initial resuscitation, mortality (50%) remains high and full recovery low (29%). Defibrillation before rewarming was successful in four of the five patients, and six of the eight victims who received Adrenaline before rewarming had a normal response rate to the administration of this drug. There were no statistically significant differences in the presence of a shockable rhythm and the success of defibrillation between the survivors and non-survivors. There was no significant difference in CPR duration between the survivors and non-survivors, despite a trend to shorter CPR duration in the survivors, suggesting that a rapid ROSC may be a positive outcome predictor.

The ERC guidelines from 2005, 2010, 2015, and 2021 indicate that “if ventricular fibrillation persists after three shocks, further attempts of defibrillation should be delayed until the core temperature levels >30 °C” [7,11,12,13]. The temperature at which defibrillation should be started, and how often it should be tried in the severely hypothermic patient, has not been established. Adrenaline should be withheld if core temperature is <30 °C [7]. The guidelines of the American Heart Association have not been updated since 2010. They state that it may be reasonable to perform further defibrillation attempts according to the standard BLS algorithm and to consider administration of a vasopressor during cardiac arrest according to the standard ACLS algorithm [8]. The Wilderness Medical Society Practice Guidelines recommend a single shock at maximal power if core temperature is lower than 30 °C [14,15]. The data from our patients come from European countries, and the accidents happened between 1994 and 2015. A maximum of four shocks was delivered in one patient. In another patient, defibrillation was withheld after one shock, despite the fact that it was unsuccessful. Adrenaline was used in eight patients, while the European guidelines suggest to not use Adrenaline before the core temperature reaches 30 °C. The difference between the guidelines and reality can be explained by the fact that accidental hypothermia is rare, that many physicians are not often exposed to such cases, and that they follow the standard guidelines for resuscitation.

Of the seven non-survivors, six received Adrenaline, and five of these non- survivors had a normal response on Adrenaline administration. Adrenaline increased the coronary perfusion pressure and ROSC after defibrillation in hypothermic dogs [14] and pigs, but not survival [10,12]. When core temperature is less than 28 °C, the efficacy of Adrenaline is reduced and accumulation can happen [3]. Unfavorable neurological outcome has been described after larger doses of Adrenaline, and frostbites can be worsened by the peripheral vasoconstriction [3]. Other studies confirm that the administration of adrenaline is a predictor of a poor outcome [16,17].

Sustained ROSC after defibrillation at <28 °C has been described, however most attempts were unsuccessful [3]. Successful defibrillation in four patients with hypothermic cardiac arrest was reported by Mair in 2019 [17]. The patients in this study had a core temperature above 24 °C, witnessed cardiac arrest, and there was immediate professional CPR with early attempts of defibrillation. Of the patients in our study, one had an unwitnessed cardiac arrest, but all four patients had core temperatures above 24 °C. A core temperature above 24 °C could be a parameter associated with successful defibrillation [17]. Optimized myocardial perfusion, can increase the chances of defibrillation under 28 °C [1,3]. Cardiac ischemia might worsen hypothermia-associated irritability of the heart [18]. Myocardial perfusion pressure can be optimized by administering vasopressor drugs [10] or by starting ECLS rewarming [1]. There is evidence that a moderate increase in core temperature during active rewarming may improve defibrillation success. Therefore, active rewarming should be started as early as possible [14].

Clinical implications are that resuscitation and defibrillation should be started immediately on-site since 14 of 88 patients in cardiac arrest attained ROSC and six of them survived with good recovery or reversible damage. Accidental hypothermia and the circumstances that lead to this pathology have high fatality rates in our cohort, which may be due to the fact that only two patients were rewarmed by ECLS. For patients at risk of CA (i.e., core body temperature < 28 °C, ventricular arrhythmia, and systolic blood pressure < 90 mmHg), rewarming should ideally take place in an ECLS center with the equipment and personnel available on site until the patient has stabilized [3]. These techniques are effective and safe, and enable a hemodynamic support and continuous oxygenation [1,19]. An on-call hypothermia coordinator, with knowledge on hypothermia for out- and in-hospital emergency services, can guide rescue services and doctors who are not often exposed to hypothermia in their decisions [20]. Other negative survival factors are unwitnessed cardiac arrest in eight, and proven asphyxia in 10 patients [6,21]. This demonstrates that even when circulation is restored, deep hypothermia has multiple pathophysiological consequences during rewarming, such as vasoplegia, decreased cardiac output, ischemia-reperfusion injury, respiratory failure, and disturbances in the cerebral and metabolic system, which can lead to multiple organ failure and death [22,23]. A total of four patients with a non-shockable rhythm (pulseless electrical activity) survived. Literature shows that PEA has a higher survival rate than asystole. On the other hand, it can be difficult to make the distinction between profound circulatory shock without ventricular standstill and PEA cardiac arrest [16,17]. Due to the subjective nature of the diagnosis of PEA, it is possible that the survivors were in pseudo-PEA (profound hypotension without ventricular standstill) and therefore responded rapidly to resuscitative measures. In addition, it can be difficult in the prehospital setting to palpate carotid and femoral pulses [17].

The main strengths of our study are the fact that it is based on the largest database of accidental hypothermia worldwide, surpassing the number of participants of other studies, which often treat just a single case of successful defibrillation before rewarming [17,24,25,26,27,28]. In addition, animal studies may not yield the same results as patients in hypothermic cardiac arrest [10].

The main limitation of this study is the low number of cases with successful pre-rewarming resuscitation. This can result in significant bias. Accidental hypothermia is rare and even with a large number of cases in the database, the number of cases with successful pre-rewarming resuscitation is low. Some cases were entered retrospectively, which can result in recall bias. Confounding factors such as the severity of trauma can influence the outcome. Another limitation is the missing data. This is normal, because a pre-hospital resuscitation is stressful and workload is high. An updated International Hypothermia Registry on REDCap has been created in 2021, which may foster better and easier data entry. Every center will have their responsible doctor, or local data manager, who adds and controls information, and follows the patient during their hospital stay.

If we compare our results with other case descriptions [25,26,29], we can see that successful defibrillation before rewarming is possible, even in older people. Other articles pose the question of whether an increase in temperature, myocardial perfusion, or the rewarming of the myocardium with warm blood in the coronary arteries makes defibrillation successful. As in the article of Kosinski 2020, we see that even without an increase in temperature, successful defibrillation is possible [26]. An increase in myocardial perfusion seems to play a key role in the success of defibrillation in moderate to deep hypothermic patients.

## 5. Conclusions

In deep hypothermic arrested patients, successful resuscitation with ROSC with a good outcome is possible before active rewarming and arrival in the hospital. Clinical implications are that resuscitation with defibrillation should be started immediately on-site because 14 of the 88 patients in cardiac arrest attained ROSC and six of them survived with good recovery or reversible damage. There were no statistically significant differences in the presence of a shockable rhythm, the success of defibrillation, and the CPR duration between the survivors and non-survivors. Other negative survival factors are that less than half of the patients had a witnessed cardiac arrest, and 10 had proven asphyxia. However, four patients with non-shockable rhythm (PEA) survived, indicating that during a rescue of hypothermic cardiac arrest patients, all efforts must be carried out to augment the chance of ROSC and survival of such patients. The main limitation of our study is that the cases are not homogeneous and that due to the retrospective character of the study, some information is missing.

## Figures and Tables

**Figure 1 ijerph-19-04059-f001:**
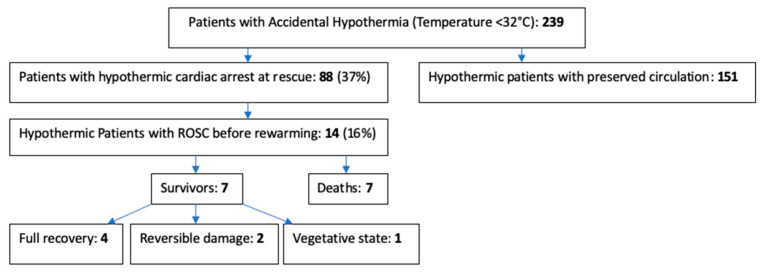
Flow chart.

**Table 1 ijerph-19-04059-t001:** Demographics of patients with hypothermic cardiac arrest and successful pre-rewarming return of spontaneous circulation. (T: core temperature, CA: cardiac arrest, U: unknown; Y: yes; N: no).

Patient	Age (Years)	Gender	T	Year of Accident	Country	Witnessed CA	Asphyxia	Mechanism	Comments
1	20	F	26.7	2012	France	U	N	Avalanche Trauma head/neck	Buried for 45 min
2	41	M	25.5	1994	France	Y	Y	Alpine	Night in a snow cave at 4600 m
3	20	M	29	2014	Austria	N	Y	Water (Submersion)	Water aspiration during 10 min
4	16	M	27.4	2011	Switzerland	N	Y	Water (Immersion) Trauma head/neck, chest	
5	25	M	25.3	2009	Switzerland	N	Y	Water (Submersion) Alcohol	
6	57	M	27	2011	Switzerland	N	Y	Avalanche	Buried for 45 min
7	28	F	29.2	2009	Switzerland	N	Y	Water (Submersion)	
8	56	M	25	2013	Poland	Y	N	Urban-Alcohol Trauma head/neck	Found on the street in the morning, Ambient temperature: 0 °C, wind: 7–9 km/h wind
9	71	M	27	2014	Switzerland	Y	N	Water (Submersion)	
10	44	M	29.8	2014	Austria	N	Y	Avalanche	
11	16	M	25	2015	Austria	Y	N	Urban-Alcohol	Outside exposure: 10 h
12	39	M	25.8	2011	Switzerland	Y	Y	Water (Submersion) Alcohol/Narcotics	
13	15	M	25.6	2014	Austria	N	Y	Avalanche	Buried for 30 min
14	22	F	28	2015	Austria	N	Y	Water (Immersion)	Car accident in lake of 10 °C

**Table 2 ijerph-19-04059-t002:** Outcome data for survivors versus non-survivors. The one surviving patient in vegetative state has been excluded. All data were analyzed as number of cases and percentages (%), median and inter-quartile range (IQR) or mean values, and standard deviations (SD) for continuous variables as appropriate. The level of significance was set at *p* < 0.05.

Variables	Survivors	Non-Survivors	*p*
Number	6	7	
Age	24.0 (20.3; 48.3)	28.3 (18.3; 42.3)	0.77
Core Temperature	26.2 (25.1; 27.8)	27.0 (26.3; 28.3)	0.28
Witnessed Cardiac Arrest	3	1 (1 unknown)	0.55
Asphyxia	3	6	0.27
Potassium	3.1 (2.5; 3.1)	3.8 (3.2; 4.3)	0.22
Lactate	8.3 (7.9; 13.1)	10.9 (9.9; 13.9)	0.36
Shockable Rhythm	2	2	1.00
Defibrillation Performed	2	2	1.00
Defibrillation Successful	2	1	--
CPR Duration	12.0 (10.0; 15.0)	29.0 (21.0; 30.0)	0.09
Rewarming Method ECLS/Non-ECLS	2	0	0.16
Adrenaline Administration	2	6	0.20
Normal Response Rate to Adrenaline	1 (1 unknown)	5 (1 unknown)	0.08

**Table 3 ijerph-19-04059-t003:** pH, Potassium, lactate before rewarming, as well as rewarming method and ICU outcome after successful pre-rewarming resuscitation and Potassium: mmol/L. (U: unknown, V-V ECMO: Veno-venous extracorporeal membrane oxygenation).

Patient	pH	Potassium	Lactate	Rewarming Method	Complications during Rewarming	ICU Outcome
1	6.87	4	8.5	External	Brain death, vasoplegia	Death
2	U	U	U	Body cavity lavage	U	Vegetative state
3	7	4.1	4.4	External	Pulmonary oedema, vasoplegia	Fully recovered
4	7.05	2.4	9.8	External, invasive CoolGard	No	Death (brain oedema)
5	U	3.1	8.3	External, invasive CoolGard	No	Survived with potentially reversible injury/pathology
6	U	3.8	16	Invasive CoolGard	Brain death	Death
7	6.75	2.7	U	Invasive CoolGard	No	Death (persistent neurologic central deficit)
8	7.07	2.5	7.9	ECMO, external	No	Fully recovered
9	U	2.1	13.1	Invasive CoolGard, external	No	Survived with potentially reversible injury/pathology
10	7.23	4.8	10	Invasive CoolGard	No	Death (brain death)
11	U	U	U	External (ECLS standby)	No	Full recovery
12	6.89	3.6	11.7	Body cavity lavage	Pulmonary oedema→ V-V ECMO	Death
13	6.91	4.5	14.6	External	Pulmonary oedema, Death	Death
14	7.2	3.1	38	Body cavity lavage→ V-V ECMO	Pulmonary oedema	Full recovery

**Table 4 ijerph-19-04059-t004:** The heart rhythm on arrival and after return of spontaneous circulation (ROSC), CPR duration, the number and success of the defibrillation attempts, and Adrenaline IV administered during resuscitation. (Y: yes, N: no, U: unknown).

	Initial Rhythm	CPR Duration (min)	Rhythm after ROSC	Defibrillation?-Successful?-Number of Shocks	Adrenaline (IV)	Response Rate
1	VF	U	Sinus	Y-N-1	Adrenaline	U
2	VF	U	Sinus	Y-Y-1	U	U
3	VF	5	Sinus	N	N	
4	PEA	35	U	N	Adrenaline 2.4 mg	Normal
5	PEA	12	Sinus	N	N	
6	VF	U	AF	Y-Y-2	Adrenaline 1 mg (3x) Adrenaline 0.1 mg (5x)	Normal
7	Asystole	20	Sinus	N	Adrenaline 4.5 mg Adrenaline 0.1 mg (5x)	U Normal
8	PEA	15	Sinus	N	N	
9	PEA	U	U	Y-Y-4	Adrenaline 3 mg	Normal
10	Asystole	29	Sinus	N	Adrenaline 5 mg	Normal
11	VF	35	Sinus	Y-Y-3	Adrenaline 1 mg (5x)	U
12	PEA	21	AF	N	N	N
13	Asystole	30	Sinus	N	Adrenaline 3 mg	Normal
14	PEA	10	Sinus	N	U	U

## Data Availability

The data presented in this study are available on request from the corresponding author. The data are not publicly available due to privacy reasons.

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
