# Peer review of "Successful Pre-Rewarming Resuscitation after Cardiac Arrest in Severe Hypothermia: A Retrospective Cohort Study from the International Hypothermia Registry"

_ijerph, 2022, doi:10.3390/ijerph19074059_

Round 1

Reviewer 1 Report

Dear Editor and authors

I feel satisfied with the answers of the questions raised by me and the other reviewer So I am favor for the publication of "Successful pre-rewarming resuscitation after cardiac arrest in severe hypothermia: a retrospective cohort study from the In ternational Hypothermia Registry "

Att.

Author Response

Dear reviewer, thank you very much for your revision.

Reviewer 2 Report

This is a much improved version compared to the previous one. The authors follow the reviewers' comments and suggestions.

I recommend approval for publication.

Author Response

(The authors gave the same response as above.)

Reviewer 3 Report

Dear authors,

The manuscript presented aims to retrospectively analyse the successful cases of resuscitation after pre-rewarming. To be able to do this, you use the IHR.

It is a good idea, but the manuscript needs to be revised. My comments would be:

  • English language including spelling and grammar must be revised.
  • Methods and results must be made clearer.
  • The study is centered in pre-rewarming, but also takes into consideration other actions such as adrenaline.
  • The sample information is unclear.
  • The dates of the cases were unclear.
  • The clinical information provided is neither clear nor homogeneous and therefore any results obtained may not be sustained by the study.

It is a good work, but needs more explanations and better patient information.

Thank you.

Author Response

Dear reviewer, thank you very much for your revision. 

  • English language including spelling and grammar must be revised.

Dear reviewer, Melodie Kaeser, a native English speaker working in the medical department, controlled the English language including spelling and grammar.

  • Methods and results must be made clearer.

Dear reviewer, we added in the method section:

73-80: The demographical data observed were: gender, age, the year and country of the accident. Prehospital data were core temperature at arrival of the rescue team, the presence of a witnessed cardiac arrest, asphyxia, the accident mechanism, the initial heart rhythm, the duration of CPR, the rhythm after ROSC. In addition, the number of successful defibrillation attempts and the reaction on Adrenaline administration before rewarming were recorded. Hospital data were the pH, potassium and lactate before rewarming (the first analysis available after arrival in the hospital), the rewarming methods and the complications during rewarming. The ICU outcome was registered.

We provided more information in the results section by:

  • Adding the date and the country of the accident in Table 1
  • Adding additional information about the accident (Table 1)
  • Adding the energy level (Joules) for defibrillation
  • The study is centered in pre-rewarming, but also takes into consideration other actions such as adrenaline.

Correct. We looked after the administration of Adrenaline pre- rewarming, in patients with a core temperature lower than 30°C, because the ERC guidelines state that Adrenaline should be withheld if core temperature is <30°C, while guidelines of the American Heart Association consider administration of a vasopressor during cardiac arrest according to the standard ACLS algorithm. The Adrenaline that was administered was pre-rewarming, in patients with a core temperature under 30°C. We specified this in the text.

51-3: Secondary outcomes were the characteristics of defibrillation, Adrenaline administration in patients with a core temperature under 30°C, and the duration of CPR.

161-3: Adrenaline was administered in 8 patients with a core temperature under 30°C, and a normal response (i.e. the same reaction as in a non- hypothermic patient) was reported in 6 patients (see Table 4).

  • The sample information is unclear.

Dear reviewer, we are sorry but we don’t understand which extra information you want about the sample. We improved the information about the sample. In Table 1, we added the year of the accident, the country and we added some more details about the accident.

  • The dates of the cases were unclear.

We added the dates of the cases. We don’t like to publish the exact date of the accident for privacy reasons, but we added the year of the accident in Table 1.

  • The clinical information provided is neither clear nor homogeneous and therefore any results obtained may not be sustained by the study.

Dear reviewer, we are sorry but can you please specify which clinical information isn’t clear for you? We hope that we gave a bit more information by adding more information about the accident. But we know that it is a limitation of our study that some information is lacking and therefore we wrote in the limitations:

251-6: The main limitation of this study is the low number of cases with successful pre-rewarming resuscitation. This can result in significant bias. Accidental hypothermia is rare and even with a large number of cases in the database, the number of cases with successful prerewarming resuscitation is low. Some cases were entered retrospectively, which can result in recall bias. Confounding factors like the severity of trauma can influence the outcome. Another limitation are the missing data.

We added in the conclusion:

279-81: The main limitation of our study is that the cases aren’t homogeneous and that due to the retrospective character of the study, some information is missing.

It is a good work, but needs more explanations and better patient information.

Thank you.

            Dear Reviewer, thank you very much. We hope that we answered your questions.

Round 2

Reviewer 3 Report

Dear authors,

Thank you for the revised document.

There are still some minor English language grammar issues, but the text is understandable.

Congratulations.

This manuscript is a resubmission of an earlier submission. The following is a list of the peer review reports and author responses from that submission.

Round 1

Reviewer 1 Report

General comments:

Thank you for opportunity for reviewing this interesting paper. Research partially adheres to STROBE guidelines.

I believe that this manuscript doesn´t qualify for acceptance at this time and should be improved for publication in IJERPH.

Specific comments:

  1. Writing

The writing, structure and organization of the manuscript is in accordance with the guidelines.

  1. Title

The title reflects the content and problem studied.

  1. Key Words

The keywords are representative of the subject studied and exposed.

  1. Background

The background reflects the state of the art in relation to the study. The objective of the study is mentioned, as well as the justification for the choice and importance of studying this theme.

  1. Methods

There isn´t detailed description of the research methods used. The design is incomplete and it isn´t possible to validate the veracity of the results

They don´t describe the setting, locations, and relevant dates, including periods of recruitment, exposure, follow-up, and data collection.

Eligibility criteria, sources and methods of selection of participants are not stated.

the explanation of statistical methods should be improved!!

Authors don´t describe any efforts to address potential sources of bias

  1. Findings

The results shown aren´t concrete and detailed,

The aim is not achievable with descriptive analysis of the data.

  1. Discussion

it not includes the main strengths and weaknesses in relation to other studies, discussing important differences in the results.

  1. Conclusion

the methodology and results do not justify the conclusion.

Reviewer 2 Report

It is a study on pre-warming resuscitation based on an international registry. Besides, it is an original topic, at least for specialties other than anesthesiology and ICU doctors.

This manuscript is well written, with a good flow of reading and the conclusions are consistent with their own research questions.

Reviewer 3 Report

Dear Editor and Authors

The topic is relevant in terms of public health, as it compares approaches to the treatment of individuals with hypothermia. The paper provides a comparative analysis of cases, treatment, recuperation/deaths of patients.

The paper provides a good introduction, justifying that recurrent defibrillation can cause myocardial injury and inconsistencies in some Guilelines Despite the information on the source of the analyzed data (International Hypothermia Registry 54 (https://www.hypothermia-registry.org), it does not provide any information about the time period of the analyzed cases, or if all cases are from the beginning of the platform (2008).

I consider that the paper brought a good survey of data, and brought up some considerations on the subject, however I felt the lack of a crossing of treatment/survival data with the age group, which must be taken into account given the possibility of heart disease in older people

Best regards